# Cloud Modelling of Property-Level Flood Exposure in Megacities

**Christos Iliadis** [1,*] , **Vassilis Glenis** [1] **and Chris Kilsby** [1,2]

1   School of Engineering, Newcastle University, Newcastle upon Tyne NE1 7RU, UK;
    vassilis.glenis@newcastle.ac.uk (V.G.); chris.kilsby@newcastle.ac.uk (C.K.)
2   Willis Research Network, 51 Lime St., London EC3M 7DQ, UK
*   Correspondence: c.iliadis2@newcastle.ac.uk

**Abstract:** Surface water flood risk is projected to increase worldwide due to the growth of cities as well as the frequency of extreme rainfall events. Flood risk modelling at high resolution in megacities is now feasible due to the advent of high spatial resolution terrain data, fast and accurate hydrodynamic models, and the power of cloud computing platforms. Analysing the flood exposure of urban features in these cities during multiple storm events is essential to understanding flood risk for insurance and planning and ultimately for designing resilient solutions. This study focuses on London, UK, a sprawling megacity that has experienced damaging floods in the last few years. The analysis highlights the key role of accurate digital terrain models (DTMs) in hydrodynamic models. Flood exposure at individual building level is evaluated using the outputs from the CityCAT model driven by a range of design storms of different magnitudes, including validation with observations of a real storm event that hit London on the 12 July 2021. Overall, a novel demonstration is presented of how cloud-based flood modelling can be used to inform exposure insurance and flood resilience in cities of any size worldwide, and a specification is presented of what datasets are needed to achieve this aim.

**Keywords:** flood risk; pluvial floods; cloud computing; flood modelling; hydrodynamic model; CityCAT; digital terrain model

## 1. Introduction

Surface water flooding is emerging as a major natural hazard due to the growth of urbanisation and the upcoming climate change that leads to more frequent flash floods from severe rainfall events in urban areas and catchments resulting in economic damage to infrastructure, assets, properties, and people worldwide [1–4]. In megacities, there is an especially notable increase in risk through anthropogenic activities increasing vulnerability [5,6]. The capacity of the current drainage system of most cities is overwhelmed during intense rainfall [7] with subsequent damage to property, critical infrastructure, and the population.

In the face of climate change and urbanisation, flood risk management is pivotal to offering adaptation solutions, and flood models are crucial to informing resilience planning in urban areas. Over the years, several research models have been developed to model drainage systems [8,9] and solve the full shallow water equations (SWEs) [10–12]. Many reviews have been written to evaluate the advantages and limitations of hydrodynamic models [13–18]. Among the many hydrodynamic models developed to solve full 2D SWEs, the City Catchment Analysis Tool (CityCAT) was employed in this study. CityCAT has undergone testing in various real flood events in the past [19–26], encompassing different cities within the UK. Additionally, it has been applied in studies conducted in the USA, with particular focus on urban flooding [27] (source: https://storymaps.arcgis.com/stories/ 3d982b40189c42aa9af56d52548caaf0, accessed on 14 September 2023). Furthermore, the

model was employed in a recent study in Greece by Iliadis et al. [28]. The accuracy and quality of the results are now sufficient to take on assessments of important locations such as megacities, where the greatest risk and vulnerabilities are found. Such large-scale modelling requires correspondingly large computational resources, and as the power of cloud computing has increased, a few attempts have been made to assess flood risk in larger cities using hydrodynamic models. Many studies simulating surface flow have been conducted for small urban catchments [29–38]. Other larger-scale studies have focused on the flood risk from rivers (fluvial flooding) [39–42]. The first attempt to simulate the flood impacts in European cities was presented by Guerreiro et al. [43], where they calculated the percentage of urban areas flooded for 571 cities in Europe with a spatial resolution DTM of 25 × 25 m for nine different rainfall events, but concluded that the low resolution of the DTM imposed major limitations due to not representing flow paths accurately. Another study evaluated flood risk by simulating the pluvial flood distribution caused during three extreme rainfall events in Shanghai with a DEM of similar (30 m) resolution [44].

Digital elevation models (DEMs) and digital terrain models (DTMs) play a key role in hydrodynamic models' production of accurate results by defining the water flow paths and flood risk in urban areas where the topography is complex due to high building and road density [45]. Xafoulis et al. [46] investigated the influence of different spatial resolutions in DEMs on flood risk assessment, focussing specifically on fluvial flooding in an agricultural region located in Greece. In terms of urban environments, recent studies by Wang et al. [47] and Jamali et al. [48] highlighted the importance of high-accuracy DEMs in flood modelling for urban flood management options through two different case studies with the use of a 1 m resolution DEM. Escobar-Silva et al. [49] explored the influence of spatial resolution in flood modelling by comparing three different rainfall events in São Caetano do Sul, São Paulo, Brazil and validated the results with field measurements provided from the local civil defence agents of the area.

This study therefore aims to investigate the limits of a high-resolution cloud-based hydrodynamic model's ability to estimate flood risk and exposure at individual building level for a large city. The critical role of DTM resolution in accuracy and runtime is established using four different grid resolutions for multiple storm depths. While performance is mostly assessed through model intercomparison, the underlying model fidelity is established with validation against field measurements from a real storm event. The demonstration of large area, high-resolution modelling and exposure analysis provides timings and costs of cloud simulations, which can be used to guide and set new standards for industry practice.

## 2. Methodology

### 2.1. Hydrodynamic Modelling with CityCAT

The City Catchment Analysis Tool (CityCAT) is a fully 1D/2D coupled hydrodynamic model, developed at Newcastle University, that can be used for modelling, analysis, and visualisation of surface water flooding [11] and urban drainage [21]. CityCAT contains explicit numerical solutions to the full shallow water equations (SWEs) [50] solved by finite volumes with shock-capturing schemes, which can handle discontinuous flows [51]. The shallow water equations can be written as follows:

$$\partial_t Q + \partial_x F(Q) + \partial_y G(Q) = S(Q) \tag{1}$$

$Q$ is the conserved quantities vector; $F$, $G$ are the flux vectors; and $S$ is the source terms vector.

The vectors are given as follows:

$$Q \equiv [q_1, q_2, q_3]^T = \left[h, hv_x, hv_y\right]^T; \; F(Q) \equiv [f_1, f_2, f_3]^T = \left[hv_x, hv_x^2 + gh^2/2, hv_x v_y\right]^T$$

$$G(Q) \equiv [g_1, g_2, g_3]^T = \left[hv_y, hv_x v_y, hv_y^2 + gh^2/2\right]^T; \; S(Q) = R - L + S_o - S_f \tag{2}$$

where $v_x$ and $v_y$ represent the depth-averaged velocity components in the $x$ and $y$ directions, respectively; $h$ is the water depth; $g$ is the gravity acceleration.

$R = [R, 0, 0]^T$ is the rainfall intensity; $L = [L, 0, 0]^T$ is the infiltration rate;

$S_o = [0, gh\partial_x z_b, gh\partial_y z_b]^T$ is the bed slope source term and $z_b$ denotes the bed elevation;

$S_f = \left[0, ghSf_x, ghSf_y\right]^T$ is the friction term (see full description of the equations in Glenis et al. [11]).

The model represents built-up areas with explicit representation of buildings by using the "*Building Hole*" approach [26], bridges [52], and different types of blue–green adaptation solutions [25]. The produced outputs of CityCAT are time series of water depth, velocity flow, flood maps and volume in and out of manholes, gully drains, buildings, etc. [53]. The minimum required inputs to simulate a study area with CityCAT are (a) digital terrain models (DTM); (b) the buildings' footprints; (c) the permeable (green) areas; (d) the rainfall intensity; and (e) the drainage system; Figure 1 highlights the steps to set up a simple simulation in an urban area.

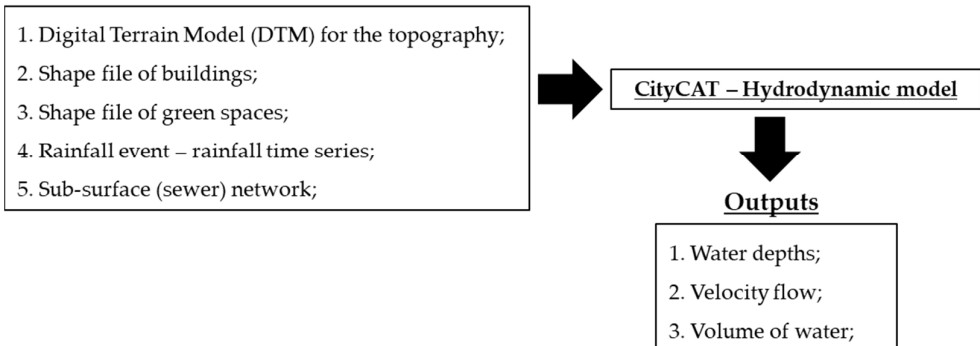

**Figure 1.** Schematic workflow to set up a simulation with CityCAT in an urban area.

### 2.2. Cloud Computing

The design of optimal and efficient solutions for flood risk management is restricted due to the limitations of combining high-performance computing with flood models. The evolution of cloud flood modelling in the last years has offered a range of options to process and store data to understand and explore flood risk management in big urban areas and catchments [19]. In most cases, use of the cloud meets specific payment restrictions for the time of renting the resources and the required random-access memory (RAM). Alternatives to the cloud usually involve a dedicated computer server, with proportional cost.

Many studies have explored and reviewed the use of the cloud for different cases, such as flood modelling, flood mapping, etc. [16,18,54–59]. A "*blade*" server installed and located at Newcastle University for research purposes is presented here and compared with the use of a cloud platform with extra payment options like Microsoft Azure.

### 2.3. LiDAR Data

Digital terrain models (DTMs) are the most fundamental input for a hydrodynamic model, as they define the computational grid and main flow characteristics. The key consideration for selection of DEM resolution is the trade-off between accuracy of flow path representation, affected by buildings as well as slopes, and speed of simulation, as a doubling in grid resolution (e.g., from 2 to 1 m) may increase run times by a factor of eight due to the reduction in time step and increase in the number of calculations and memory requirements. Validation against historic storms in the past shows that 1 m and 2 m grid squares satisfactorily resolve streets and other flow paths between buildings while grid squares of size > 5 m may close flow paths between buildings, resulting in unrealistic flood depths [20,26].

For the UK, LiDAR derived DEM data is available from Digimap (Digimap (edina.ac.uk, accessed on 15 February 2022)) at different resolutions with unit pixels in metres. This study explores the influence of high-resolution DTMs in megacity flood modelling and the RAM required for the simulations. The resolutions of the DTMs used in this case study are 1 m, 2 m, 5 m, and 10 m. Table 1 shows an example of computational grid squares with the RAM required to run a simulation with CityCAT.

**Table 1.** Number of cells in a computational grid and required memory to run CityCAT model.

| Number of Cells in a Computational Grid | Required RAM in GB (Approximate) |
|:---:|:---:|
| 500,000 | 16 |
| 1,500,000 | 20 |
| 10,000,000 | 40 |
| 15,000,000 | 60 |
| 50,000,000 | 200 |

*2.4. Estimating Flood Exposure to Buildings*

The flood exposure tool, initially developed by Bertsch, Glenis [24], was used in this work to estimate the flood risk to buildings and classify them according to the water depth in a buffer zone with a simple scheme (see Table 2). The mean and the 90th percentile of water depth were extracted for each building of the study areas in multiple buffer zones around the building perimeter. Note that the buffer zone depends on the DTM resolution (the proposed buffer zones are 1.50 m for the DTM with a 1 m resolution, 3 m for the DTM with a 2 m resolution, 5 m for the DTM with a 5 m resolution, and 10 m for the DTM with a 10 m resolution). These depths can be used for damage estimation using depth–damage curves as well as a classification. A threshold of 30 cm was used to classify the buildings according to flood risk.

**Table 2.** Classification criteria to calculate the flood risk likelihood to buildings.

| Exposure Class | Mean Depth (m) | 90th Percentile (m) |
|:---:|:---:|:---:|
| Low | <0.10 | <0.30 |
| Medium | <0.10 | ≥0.30 |
| | ≥0.10–<0.30 | <0.30 |
| High | ≥0.10 | ≥0.30 |

*2.5. Rainfall Data*

The FEH22 rainfall depth–duration–frequency (DDF) model was used with the latest rainfall estimation for the area of central London—Piccadilly Circus [60] from the UK Centre for Ecology and Hydrology [61]. The storm profiles were generated following the FEH rainfall–runoff method [62]. Table 3 presents the storm events for multiple return periods for a 1 h duration; among them is the historic storm event that hit London on the 12th of July 2021 with 76.20 mm of rainfall within 90 min. This extreme event corresponds to a 1 in 484-year return period and was used to validate the observed data with the modelled output. The intensity of precipitation for this event was more than twice the average July total rainfall for London in less than two hours. Figure 2 shows the generated storm profiles for the range of return periods. A full risk assessment should consider storms of multiple durations as well as multiple return periods (depths) to establish the overall risk, which may vary across the domain, as different areas will have different catchment sizes and therefore different critical durations. A comprehensive coverage of durations and return periods was not possible in this study due to computational and time constraints, so a single duration was selected for ease of analysis and comparison with other studies. Storm

events of one hour were used for this initial study, as the effective average catchment size for London is relatively small (of order 10 km$^2$) and the majority of flooding in recent years was caused by events of around one hour long.

**Table 3.** Storm event depths for multiple return periods.

| Return Period | Rainfall (mm) |
| --- | --- |
| 2 | 11.7 |
| 5 | 20.4 |
| 10 | 26.7 |
| 20 | 32.9 |
| 50 | 41.5 |
| 100 | 48.4 |

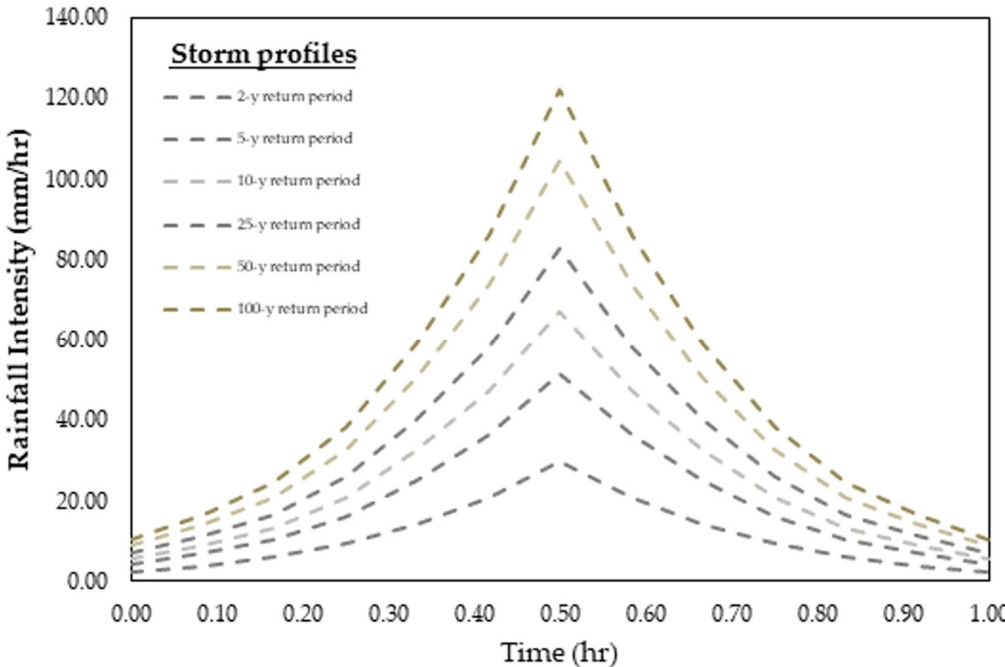

**Figure 2.** Storm profiles for multiple return periods with a 1 h duration for the area of London.

## 3. Area of Interest and Modelling Setup

### 3.1. Case Study

The primary analysis focuses on a part of the Lea catchment, Central London, UK with an area of 37.6 km$^2$, which is subject to major flood risk. This catchment has been hit by severe storm events in the last decade, twice recently in July 2021, resulting in damage from surface flooding to many houses, basements, businesses, and underground stations as reported by the Mayor of London [63]. Moreover, this study examined flood risk during multiple storm events and a range of DTM resolutions for the City of London, Westminster, Kensington, and Chelsea, where historic buildings are located, such as Westminster Abbey, Big Ben, and the British Museum, as well as residential properties, commercial places, and large green spaces such as Hyde Park, Green Park, and Regent's Park. Hence, this part of London is highly exposed, with Oxford Street having more than 500,000 pedestrians per day [64], and ageing underground stations (Piccadilly Circus, Baker Street, Covent Garden, etc.). Figure 3 illustrates the study catchment for the first part of the analysis where the locations are highlighted. Moreover, a larger part of London, which covers an area of 687 km$^2$ and includes more than 1,700,000 buildings, was selected to explore the usage and

cost of the cloud for flood modelling with the CityCAT model; more details are discussed in Section 4.4.

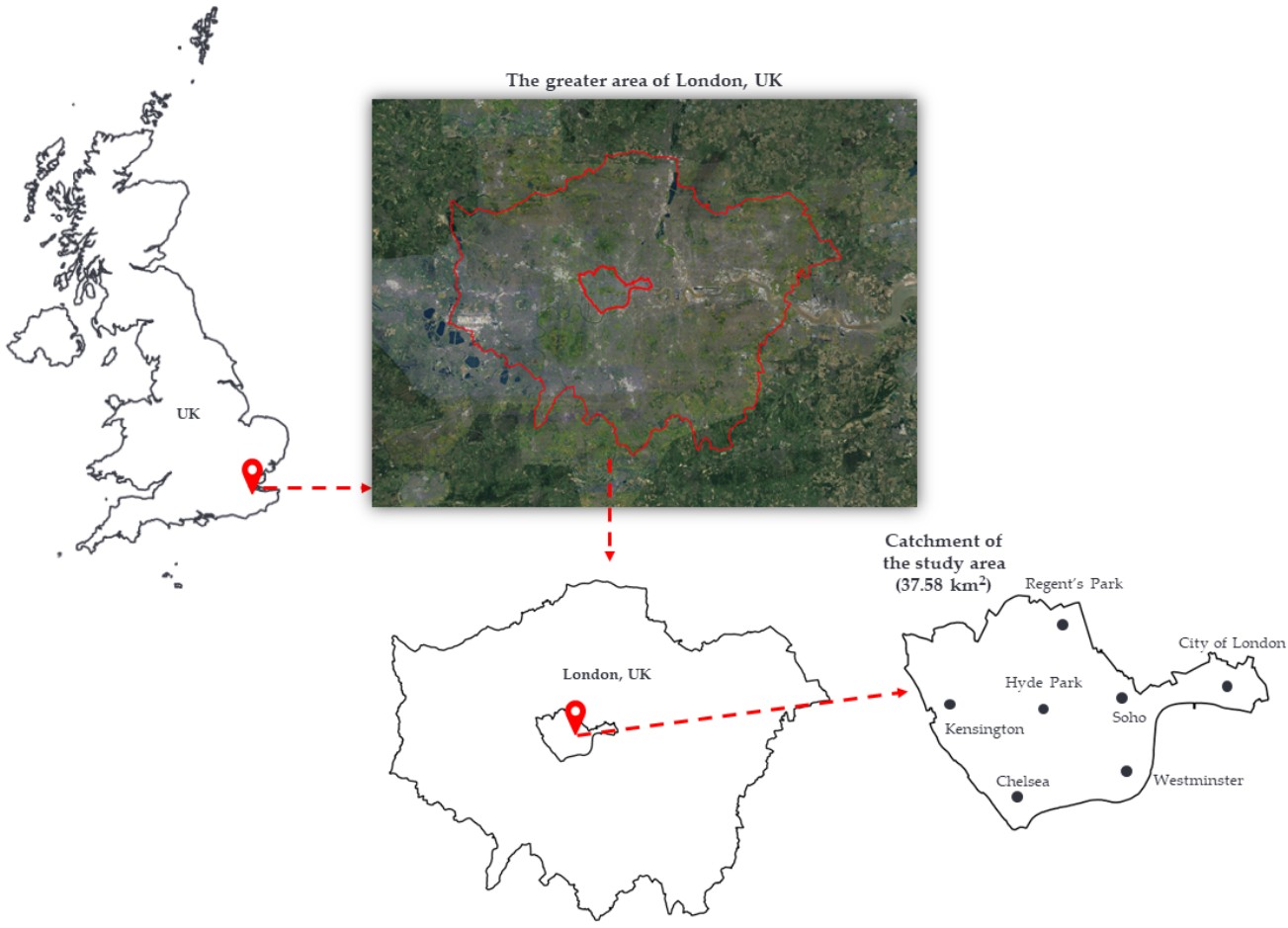

**Figure 3.** Overview of the study area in London, UK.

*3.2. CityCAT Setup*

The overland flow over and around urban features (buildings, green spaces) has been simulated using the hydrodynamic model CityCAT [11] for storm events of 60 min and return periods of 2, 5, 10, 20, 50, and 100 years. Additionally, the historic storm of July 2021 was simulated with a 1 in 484-year return period design storm with a duration of 90 min. Simulations were carried out for multiple spatial resolutions of the DTMs, e.g., grid squares of area 1 m$^2$, 4 m$^2$, 25 m$^2$, and 100 m$^2$. The buildings and the permeable areas were extracted from OS Mastermap Topography [65]. The "*Building Hole*" technique was used in all models for the representation of the urban features, where the buildings' footprints are removed from the computational grid and the rainfall on every roof is redistributed to the nearest surface grid square [26]. The total number of buildings in the study area is 95,976. The advantage of this approach is that offers more realistic results, which validate well against observed data from real storm events, and it is easy and simple to categorise buildings according to their flood risk as well as to calculate the damage from surface flooding. For the sake of simplicity and ease of use, the catchment boundary conditions were kept open.

The computational grid squares in the domain comprise 25,199,282 cells, 6,299,585 cells, 1,007,735 cells, and 255,786 cells for the DTMs with resolutions of 1 m, 2 m, 5 m, and 10, respectively. The Green–Ampt method was used to calculate the infiltration of water in permeable areas [66]. A significant limitation to this study is that the sewer network was excluded from all simulations due to the limited available data. While some practitioners make an allowance for this by reducing the input rainfall by, e.g., 20 mm (see Iliadis

et al. [28]), for transparency and intercomparison, we did not apply any correction. An alternative option would be to decrease the rainfall intensity to match the intensity associated with the concentration time derived from the intensity–duration–frequency curve corresponding to the design frequency, as per standards in place when the sewage system was originally commissioned. For the largest storms simulated here, it is expected that the storm sewer system would overwhelmed in any case, as it is, in principle, only designed to drain an up to roughly 20-year return period storm event. All the simulations were performed on the Newcastle University blade server with 767 GB of RAM memory, except the simulation for Greater London, where the Microsoft Azure platform was used. Table 4 shows the required memory and run time for every simulation per rainfall scenario.

**Table 4.** Number of grid squares, required RAM, and simulation time per storm scenario.

| Number of Cells in Computational Grids | Cell Size | Required RAM (GB) | Simulation Time per Storm Scenario (min) |
|---|---|---|---|
| 255,786 | 10 m | ≈16 | 10 |
| 1,007,735 | 5 m | ≈20 | 30 |
| 6,299,585 | 2 m | ≈40 | 300 |
| 25,199,282 | 1 m | ≈122 | 1200 |

## 4. Results—Flood Risk in London

Flood risk management in megacities, like London, is a critical aspect of urban planning and is exacerbated more than in the case of normal cities by the extra vulnerabilities of large (underground and overground) mass transit networks for their larger populations. Flood modelling is also crucial to these large cities in terms of insurance exposure, as very large risk portfolios for residential and business properties are built up, requiring reinsurance to spread the risk.

### 4.1. Modelled Flow Depth

In this section, the flood depth, number of buildings exposed to flooding, water flow paths, and estimated inundated damage of each model were compared for a 1 in 100-year storm event with a 60 min duration. The complex topography, roads, and the low gradient of the surface elevation in this part of London allow for examination of the direct influence of flooding on urban features and the detailed changes to flood flow paths.

For models with lower spatial resolution (i.e., 5 and 10 m), significant underestimation of water depths, the buildings exposed to flooding and the changes to water flow paths can be seen in Figure 4. The differences between 1 m and 2 m resolutions are minor, and the main flow paths in the domain can clearly be seen. The 5 m resolution model outputs show blocking of the main roads in the catchment, and only the major flow paths associated with natural channels are satisfactorily captured. The use of 10 m resolution in the study area results in the underestimation of water depth and the occurrence of unrealistic concentrations of water in certain locations. This leads to the formation of unrealistic ponding upstream without posing a severe flood risk. An artefact of the low spatial resolution across the computational domain is the systematic differences in water depths. Figure 5 shows the distribution of modelled water depths among the different spatial resolutions of the DTMs, which shows that low resolution modelling cannot produce the full range of flooding observed. Note here that the very high depths in these tables correspond to the Thames River and several ponds in the study area.

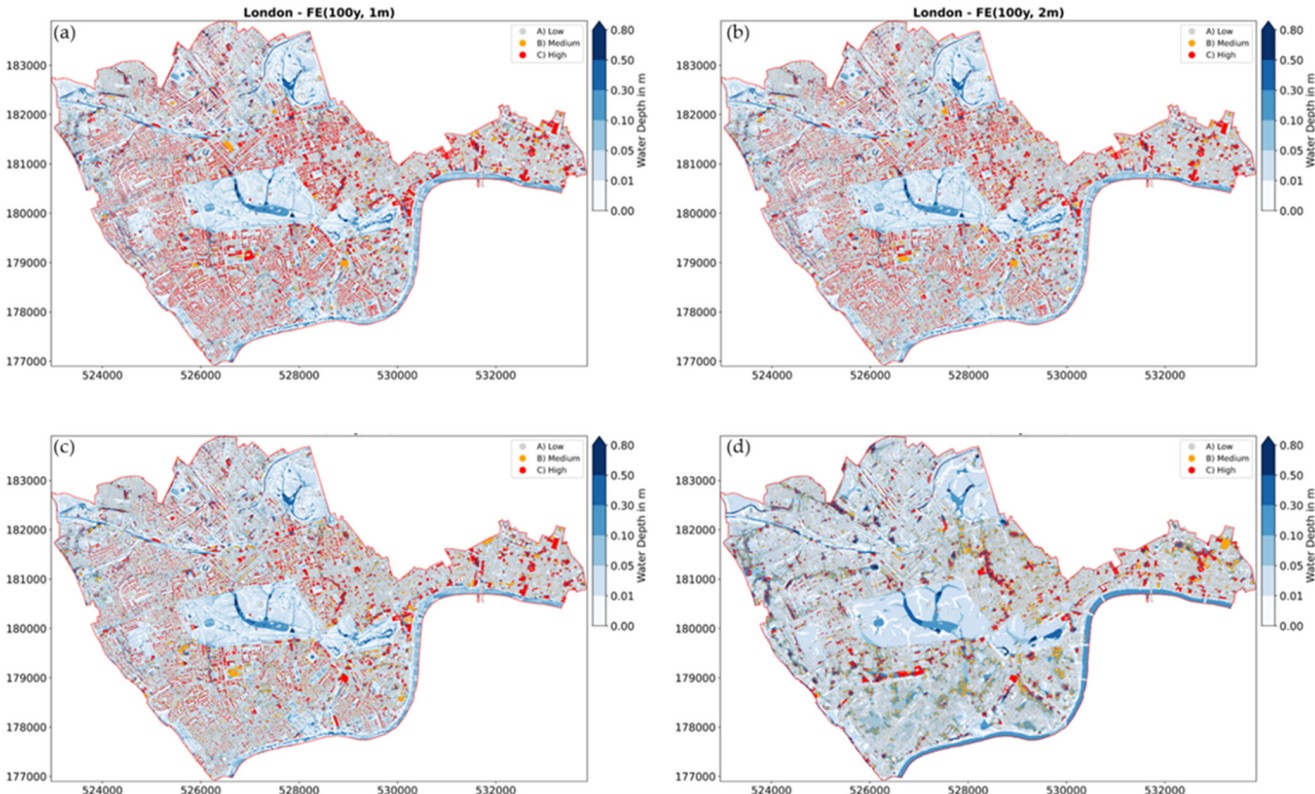

**Figure 4.** Example of flood exposure and modelled water depth for a 1 in 100-year storm event for grid resolutions of (**a**) 1 m; (**b**) 2 m; (**c**) 5 m; and (**d**) 10 m.

In general, considering the critical importance of accurate flood modelling in densely populated urban areas, use of high-resolution DTMs is crucial to achieving reliable results for flood risk management. The findings presented in this study highlight the limitations of lower spatial resolution DTMs (5 m and 10 m) in accurately simulating flood depths, identifying buildings exposed to flooding, and capturing water flow paths in the urban fabric. Such underestimations and inaccuracies in flood modelling could have serious implications for designing effective flood defences. As megacities, such as London, continue to experience rapid urbanisation and face challenges of climate change, including more intense and frequent storm events, it becomes ever more important to use high-resolution DTMs (1 m or 2 m).

### 4.2. Exposure and Flood Damages to Urban Features

To identify and compare the urban features exposed to flood risk for multiple storm scenarios and resolutions of the DTMs, the flood exposure calculator was used (see Section 2.4), developed by Bertsch et al. [24]. There are 95,976 separate unique buildings identified under MasterMap coverage in the domain. Figure 6a highlights the buildings exposed to surface flooding per storm scenario and different resolutions of the computational grid. The model with a 1 m resolution estimates the largest number of buildings at flood risk for all the intensities of rainfall and the 10 m the smallest, which is consistent with the correct capture of the water flow paths in the domain. Figure 6b presents the percentage of buildings at flood risk in the study area, where the 1 m resolution DTM again shows the highest affected buildings from inundated depth. Figure 7 displays the buildings identified as being at high flood risk for multiple DTM resolutions. It is evident from the table that there is a noticeable decrease in the number of buildings at high risk estimated using the 10 m resolution model compared with the 1 m model. Figure 8 illustrates water depths and buildings exposed to flooding in a selected area of London, Mayfair, with a total of 3430 buildings. It can be seen that for lower spatial resolution, e.g., 5 m and 10 m, this

shows an underestimation similar to that seen in the larger domain. It can be seen that the use of a low-resolution DTM (e.g., 10 m) introduces erroneous obstructions to the flow path, resulting in increased flood risk upstream while simultaneously reducing the flood risk downstream. The disparity in building assessments is shown in Table 5, which illustrates the differences in the count of buildings exposed to elevated flood risk between the 1 m DTM and the 10 m DTM. While the classification scheme is unchanged for 2029 buildings, substantial shifts are seen for the remainder, such as transition from low to high risk (e.g., 575 buildings, constituting 16.8% of the total urban features) and vice versa (e.g., 826 buildings, accounting for 24.1% of the total), rather than gradual shifts between medium and high or high and medium risk. While the total number of buildings classified as high flood risk at 10 m resolution is reduced to around half that at 1 m resolution, this change is actually the net result of 826 buildings at reduced risk (mostly downstream of blockages to the flow pathways) and 575 at increased risk (mostly upstream of blockages).

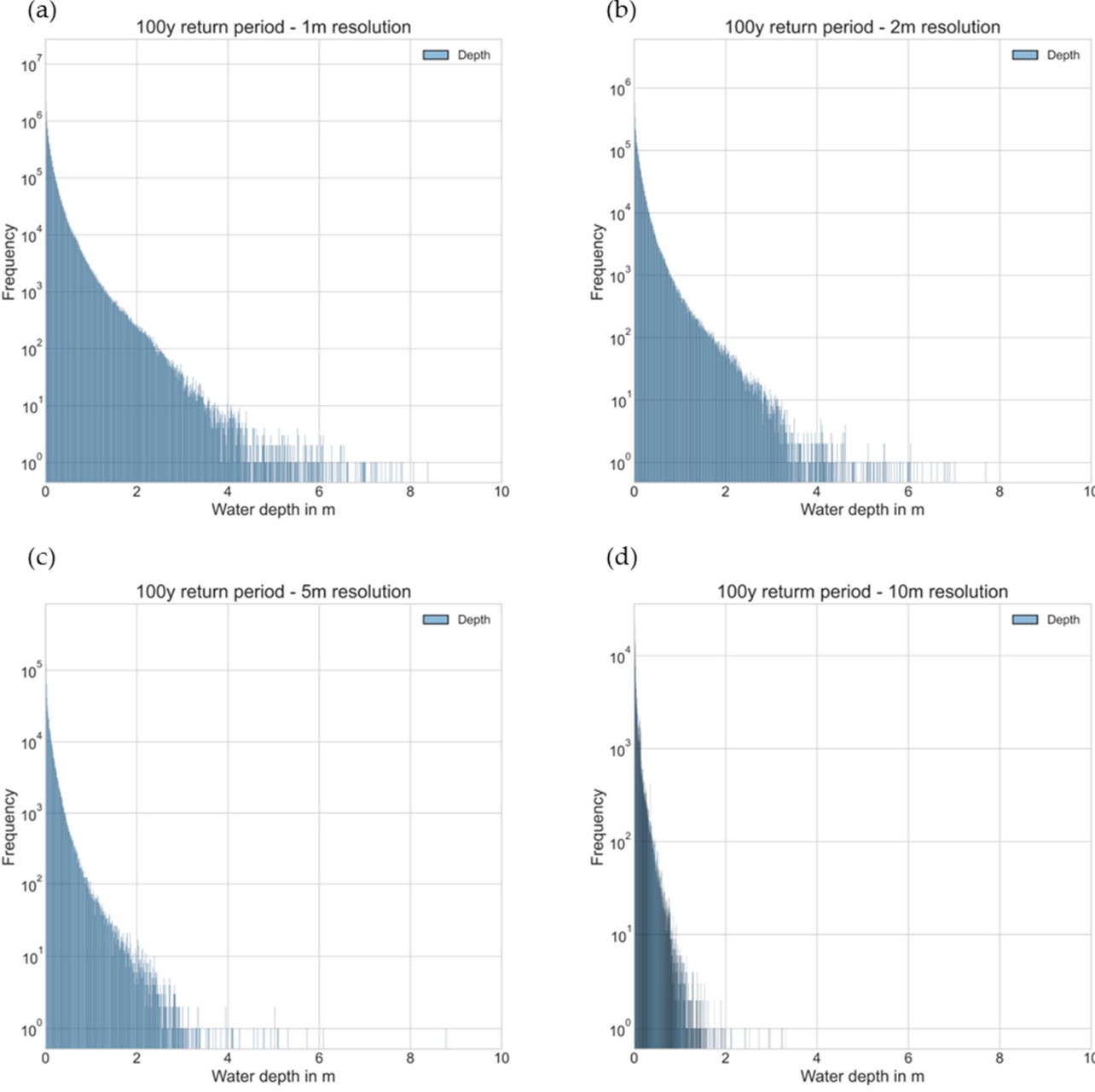

**Figure 5.** Distribution in water depth for (**a**) 1 m, (**b**) 2 m, (**c**) 5 m, and (**d**) 10 m resolutions.

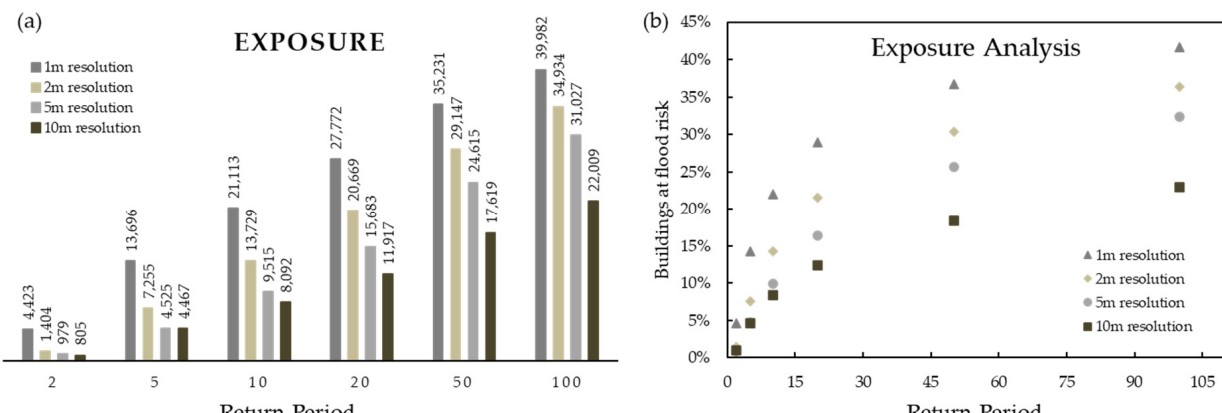

**Figure 6.** (**a**) Total numbers and (**b**) percentage of inundated buildings per storm scenario and per DTM resolution.

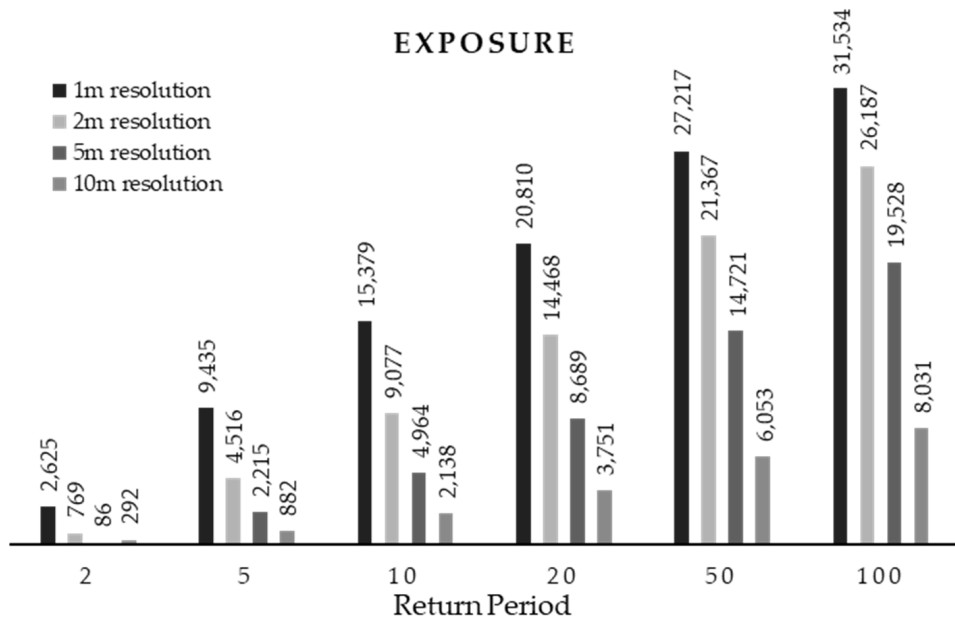

**Figure 7.** Number of buildings at high flood risk per rainfall scenario and per DTM resolution.

**Table 5.** Analysis of changes in numbers of buildings with flood risk for 1 m and 10 m DTM resolution models for a 1 in 100-year storm event.

| | Number of Buildings (Total 3430) | Percentage of Total |
|---|---|---|
| High flood risk—1 m model | 695 | 20.3% |
| High flood risk—10 m model | 363 | 10.6% |
| No change from 1 m and 10 m models | 2029 | 59.2% |
| Change: zero/low/medium to high | 575 | 16.8% |
| Change: high to zero/low/medium | 826 | 24.1% |
| Net change: high to zero/low/medium | 251 | 7.3% |

The conventional approach of calculating the estimated damage from flooding was followed here with a depth–damage curve (DDC), with the simplification that the buildings in the study area are either all residential or all commercial. In megacities, it is a very challenging task to categorise buildings (The GeoInformation Group (2014): UK building

classes; NERC Earth Observation Data Centre, https://catalogue.ceda.ac.uk/uuid/cf27c8 1e7f54c3701899017d1b810f81, accessed on 7 July 2023) according to their type or to find proper data with all these useful pieces of information. The proposed prices from the Handbook for Economic Appraisal (Multi-Coloured Handbook, Priest et al. [67]) were used to calculate the damage to residential and commercial properties. Average damages for residential and commercial buildings are shown in Figure 9. For clarity, the buildings at low risk were excluded from the damage calculation by assuming that the damage is only significant for buildings identified at medium and high risk.

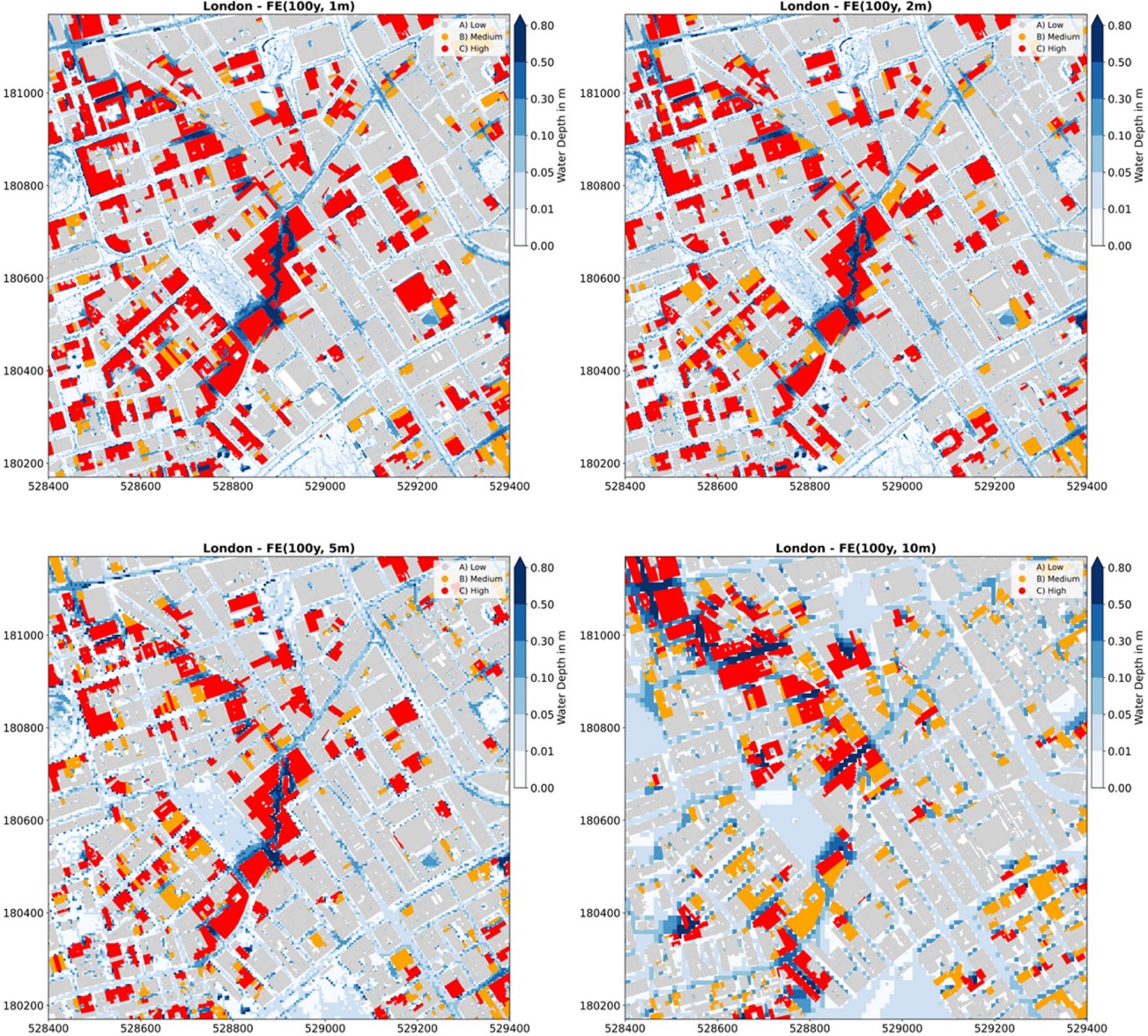

**Figure 8.** Examples of water depth and flood exposure to buildings, inundation maps for a 1 in 100-year storm event for (**upper left**) 1 m, (**upper right**) 2 m, (**lower left**) 5 m, and (**lower right**) 10 m resolutions of the computational flow domain. FE refers to flood exposure. Red, orange, and light grey colours define buildings at high, medium, and low risk, respectively, while blue shades are water depths.

The estimated total flood damage per storm scenario and per different resolution of the computational grid is presented in Figure 10. It can clearly be seen that improving the model's resolution increases the total damage successively, with a factor of three increase from 10 m to 1 m resolution, and even around 25% from 2 m to 1 m. The modelled water

depth, exposed buildings, and total estimated damages in the Mayfair area of London are shown in Figure 11. Major differences can again be seen between coarse resolution (10 m) model estimates and higher-resolution estimates (1 m and 2 m). This example shows that coarse models (10 m or worse) can substantially misidentify areas of flood risk, in this case by severely underestimating the risk in the centre of the map and overestimating the risk in the northwest sector.

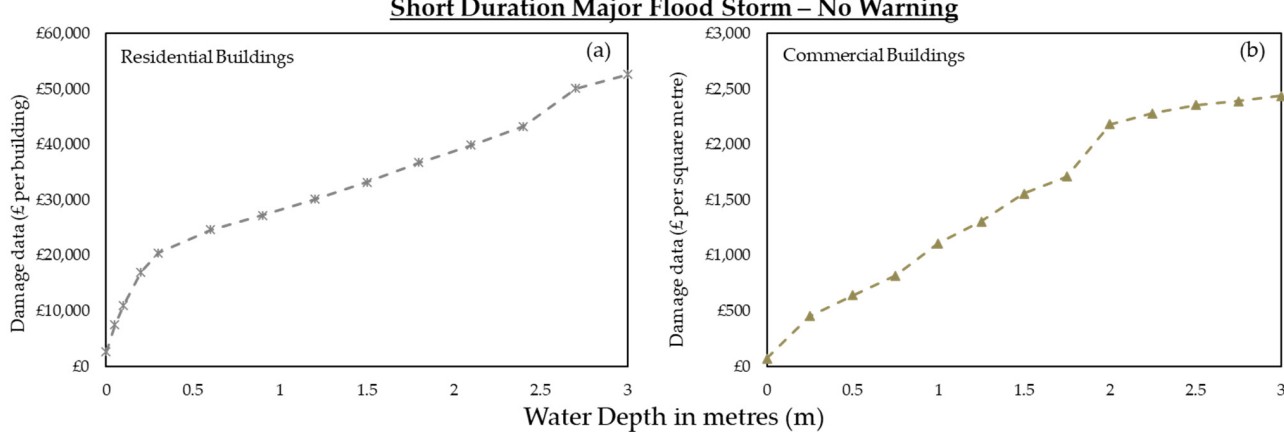

**Figure 9.** Depth–damage curves for direct damage from different water depths for (**a**) residential and (**b**) commercial buildings [67].

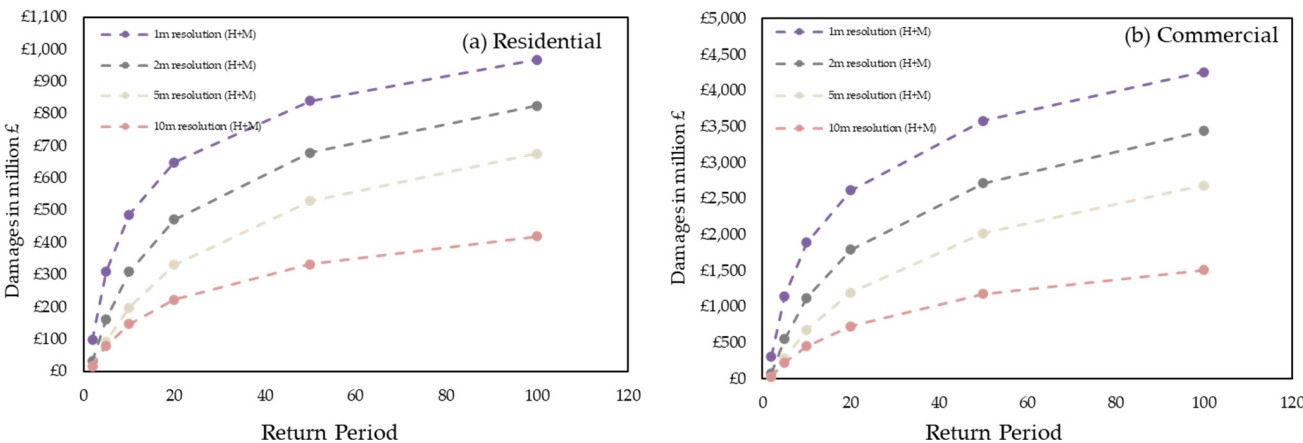

**Figure 10.** Estimated total damages per storm scenario and per spatial resolution.

### 4.3. Validation against Real Storm Event

Validation against real events is a fundamental step in assessing the reliability and accuracy of hydrodynamic models by comparing observed data from actual storm events with model predictions to increase flood resilience planning and design in cities. This process builds confidence in the model's ability to accurately simulate flood events and has largely been absent from commercial modelling of urban floods to date, but there is potential due to increased availability of flood depth data from social media and citizen science, e.g., Loftis et al. [68] and See [69]. In this section, a validation between affected locations during a real rainfall event and the outputs from CityCAT are compared. Following the extreme storm event on the 12th of July 2021, fourteen flood points across the area of London were selected (the flood points correspond to roads with buildings), where the observed depth was estimated from flood pictures downloaded from the Twitter platform during the day of this extreme event and from statements of people affected (Table 6). This comparison aims to ensure that the modelled water depth from CityCAT corresponds to the observed data. The resolution of the DTM for the validation has been chosen at 2 m, as it resolves

the water flowpaths quite well in large catchments, as discussed in Section 4.1. Table 6 presents the affected sites with the observed (Dobs refers to observed depth) and modelled data (Dmin refers to minimum model water depth and Dmax to maximum model water depth), while Figure 12 is a graphical comparison of the results. To ascertain the range of estimated water depths (simulated range) at the observed points, a 12 m buffer zone was generated to encompass the neighbouring computational cells. Both the model and the observed values are associated with the nearest grid square location. It can be seen that there is some overestimation of the depths by the model, which is consistent with the exclusion of the drainage system from the simulations (see point 12). The largest difference (at point 4) between the observed and the modelled water depth is most likely because the observed depth is measured inside the property (see Figure 13 for the fourteen flood points with the flood picture) while CityCAT excluded the buildings from the computational flow (see Section 3.2) and estimates the depth in the nearest surface grid of the building. In the other flood points, the modelled inundated depth is satisfactorily close to the observed depth. Figure 14 illustrates the likelihood of inundation exposure to buildings in the study area during this historic storm event.

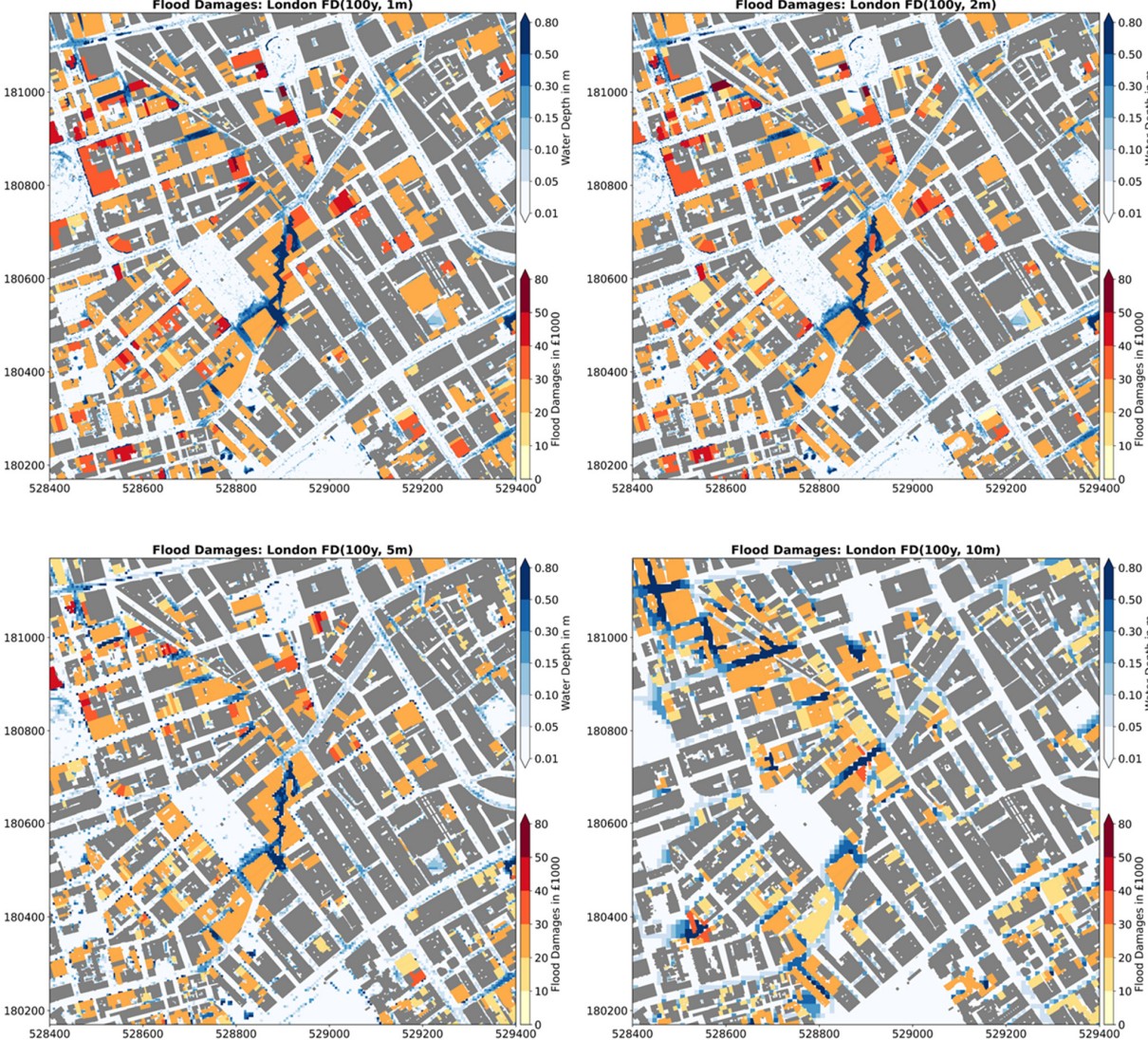

**Figure 11.** Examples of water depth, flood exposure, and damage to building maps for a 1 in 100-year storm event for (**upper left**) 1 m, (**upper right**) 2 m, (**lower left**) 5 m, and (**lower right**) 10 m resolutions of the computational flow domain. FD refers to flood damages, and yellow to red defines the cost per building from flooding.

**Table 6.** Flood validation points in London during the 2021 storm event with observed and model-estimated water depth.

| A/A | Flood Points | Dops | Dmin | Dmax | Model Depth in m |
|:---:|:---:|:---:|:---:|:---:|:---:|
| 1 | Horse Guards Road | 0.07 | 0.034 | 0.49 | 0.10 |
| 2 | Leicester Square | 0.01 | 0.002 | 0.16 | 0.02 |
| 3 | Piccadilly Circus | 0.10 | 0.002 | 0.22 | 0.13 |
| 4 | Ladbrook Grove | 0.60 | 0.001 | 0.82 | 0.82 |
| 5 | Maida Vale | 0.65 | 0.001 | 0.75 | 0.75 |
| 6 | Portobello Road | 0.38 | 0.110 | 0.46 | 0.46 |
| 7 | Dorset Square | 0.25 | 0.133 | 0.34 | 0.34 |
| 8 | Maida Vale | 0.07 | 0.116 | 0.27 | 0.08 |
| 9 | TFC Camberwell | 0.23 | 0.204 | 0.42 | 0.29 |
| 10 | Hackney Wick DLR Station | 0.22 | 0.400 | 0.61 | 0.25 |
| 11 | New Covent Garden Market | 0.30 | 0.002 | 0.34 | 0.34 |
| 12 | Brookfield Rd | 0.35 | 0.510 | 0.99 | 0.40 |
| 13 | Lea Bridge Road | 0.90 | 0.002 | 0.96 | 0.96 |
| 14 | Idea Store Whitechapel | 0.22 | 0.002 | 0.84 | 0.25 |

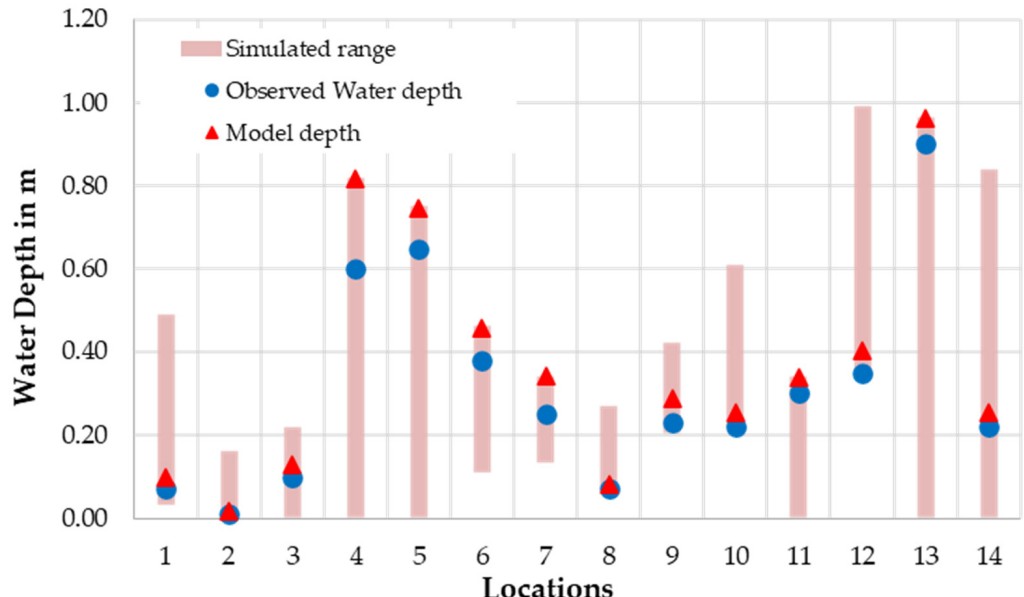

**Figure 12.** Comparison of the modelled and the observed water depths.

The CityCAT model demonstrated acceptable accuracy in predicting depths at affected areas during validation. This highlights the model's effectiveness in detecting areas that may be impacted by various factors, such as floods. It is worth noting that the modelled water depths, on average, show an overestimation of around 23%. This overestimation can be attributed to the exclusion of the drainage system from the simulations, and that there is a systematic bias due to the data observation being carried out by eye at the deepest point due to limited access to flood survey data. This validation was carried out with opportunist reports of flood depths, focussing on areas where flooding was severe. A more balanced and substantive approach in addition to comparing these "*true hits*" would more systematically consider areas where actual depths were low and model depths were high (i.e., "*false hits*") as in, for example, Bertsch et al. [24]. Such an approach requires a

systematic survey of property owners and residents, and this was not available at the time of writing.

**Figure 13.** Overview of the study area with the validation locations and modelled inundation depth for the whole domain.

### 4.4. Cloud Flood Modelling—The Greater Area of London

Assessing flood risk in megacities, like London, is always a challenging task due to the limitation of computer power and the possible high cost of using the cloud. In this section, a flood risk analysis with a DTM at a high spatial resolution of 4 m$^2$ (grid square is 2 m) for all individual properties in Greater London is presented for a range of intense rainfall events by using the power of the cloud to model an area of 687 km$^2$ that comprises 132,857,544 computational cells in the flow domain and approximately 1,750,914 buildings. Thus, this approach is suitable for a densely built up area such as London. The outputs of this analysis are at property level, so in principle, and with appropriate validation, could be appropriate for detailed insurance portfolio assessment, as well as large-scale strategic planning, resilience, and climate change stress tests.

The Microsoft Azure platform [70] was used to perform all the simulations of this area with 700 GB RAM memory and almost 20 h of CPU time for each storm event with a one hour duration. The advantage of the Azure platform is that it provides the same simulation cost per hour for all the instance types of resources and was chosen for that reason [19], with different configurations ranging from 1 core with 1 GB RAM to 96 cores with 1 TB RAM. The final cost of every simulation was around GBP 12 per hour. Calculating the likelihood of exposure to urban features for each storm event required an additional four hours per storm scenario on the Newcastle University blade server. Table 7 shows the buildings estimated to be exposed to flooding for multiple storm events for a storm with a

1 in a 100-year return period where the total urban features correspond to 16% of buildings in the study area. Figure 15 illustrates the estimated model water depth and the buildings exposed to flooding in the Greater London area (more flood exposure maps are available as Supplementary Material).

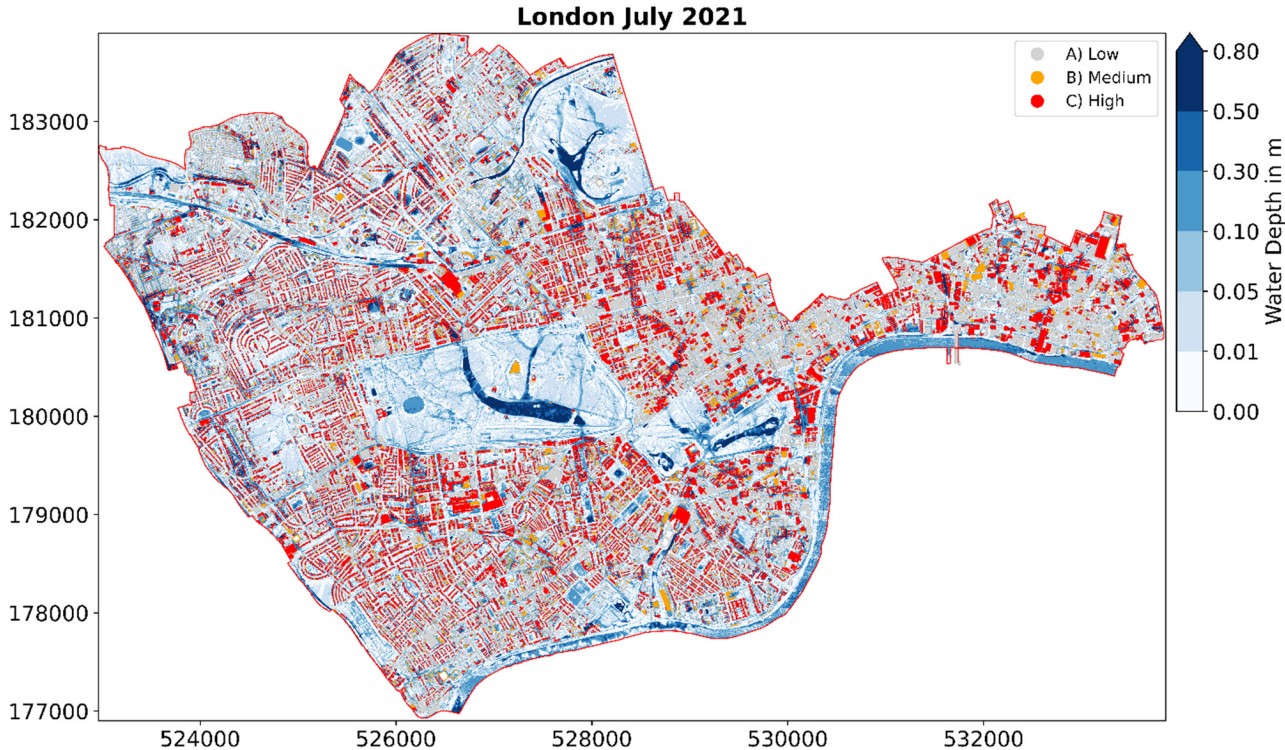

**Figure 14.** Water depth and flood exposure to buildings during the storm event in July 2021 for the central London (first part of validation).

**Table 7.** Total number of inundated buildings per storm event for the Greater London area.

| RP | Medium | High | Total |
|----|--------|------|-------|
| 2 | 5159 | 5447 | 10,606 |
| 5 | 13,458 | 15,274 | 28,732 |
| 10 | 37,553 | 48,948 | 86,501 |
| 20 | 50,414 | 68,337 | 118,751 |
| 50 | 63,189 | 89,885 | 153,074 |
| 100 | 105,381 | 163,516 | 268,897 |

This is the first time that such a large urban area has been modelled with a hydrodynamic model at such a high spatial resolution and for a range of storm events. The industry standard until now for large areas has typically used a DTM at 5 m resolution and the "stubby" platform for the representation of buildings where this approach, according to Iliadis et al. [26], causes unrealistic water flow paths in the domain and systematically underestimates flood risk. This study is a clear demonstration that modern, efficient codes like CityCAT, coupled with cloud-based computing, obviate the need for simulating large domains at either inadequately low resolutions or with inefficient subdivisions of the domain.

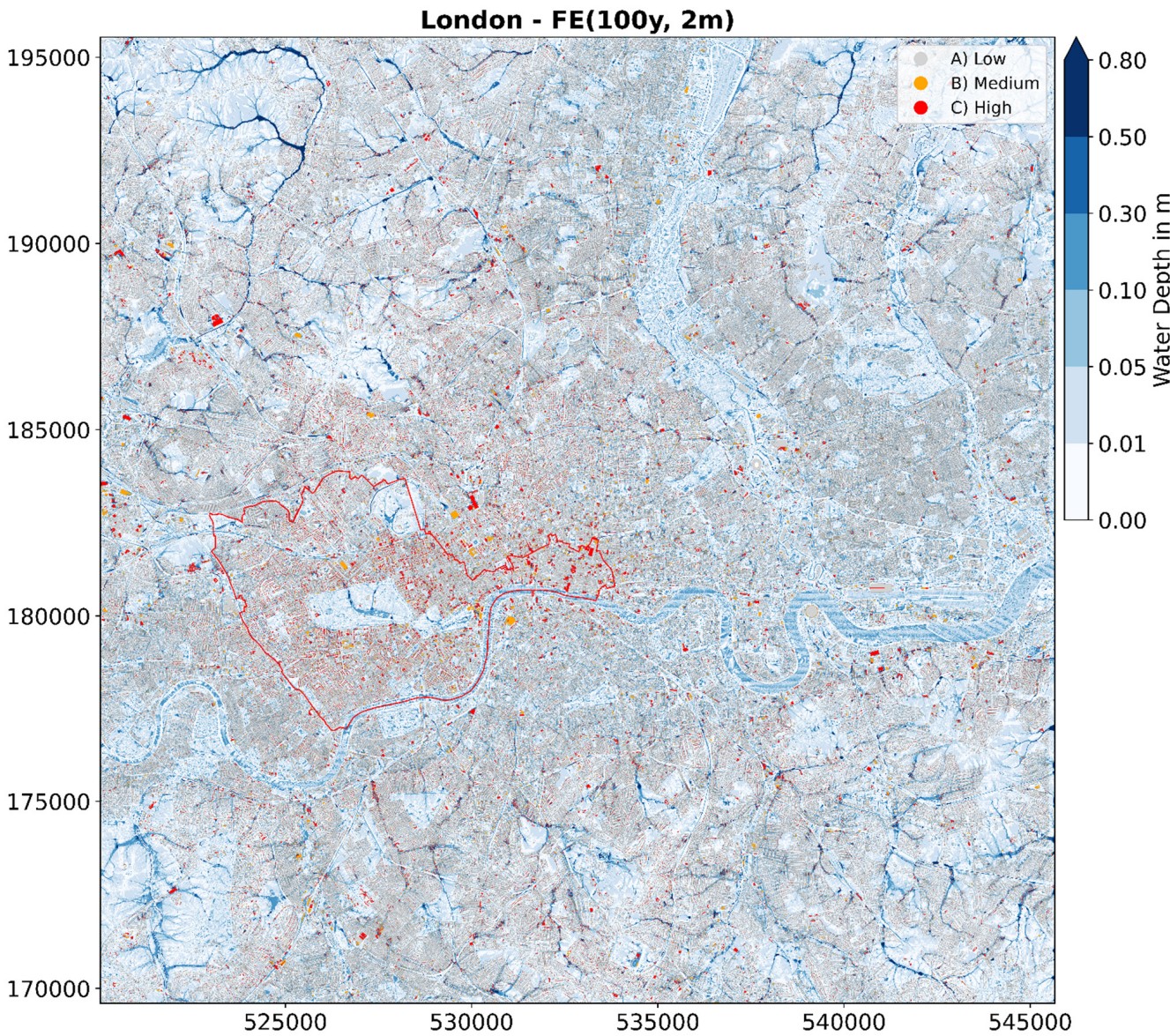

**Figure 15.** Example of the modelled domain of Greater London. Flood depths from CityCAT simulation and flood exposure to buildings for a storm event for a 1 in 100-year return period with one hour duration. The red colour defines buildings at high risk, the orange at medium risk, and light grey at low risk.

## 5. Discussion and Conclusions

This study illustrated the critical role of DTM resolution in large-scale hydrodynamic flood modelling using an application that evaluates the flood exposure to individual buildings in a large city. The high-resolution hydrodynamic model CityCAT, operating on the Azure platform (cloud), was presented to assess flood risk in megacities, which provides a template and guide for modellers engaged by insurers, local authorities, and other risk managers and planners to define modern assessment strategies and workflows.

Water flow paths and flood depths were well captured with high spatial resolution DTMs, such as at 1 m and 2 m resolution, while with lower resolution DTMs (5 m or more), many flow paths are systematically blocked due to buildings. In many cases, with low resolution DTM models, blocked flow paths lead to some overestimation of water depths upstream, while widespread underestimation occurs downstream, leading to unrealistic results due to falsely highlighting areas as high risk. Moreover, assessing the exposure flooding likelihood of urban features at high resolution offers more accuracy in identifying

and locating all the exposed buildings, in contrast with low-resolution modelling where there is overall manifest underestimation.

A validation of model estimates of water depth during a real storm event in multiple places in London showed that the use of a 2 m resolution DTM in CityCAT successfully predicts the water depth, with an overestimation of 23% consistent with the exclusion of the sewer system from the simulations and systematic bias by eye. A more comprehensive and systematic validation is planned when flood survey data are available. Overall, the model results show good correlation with observed flood data from a major pluvial flood on 12 July 2021.

Finally, cloud computing has enabled higher-resolution pluvial flood modelling and access to enough resources to allow for simulations of multiple storm events in larger areas than before with the hydrodynamic model CityCAT. The novel city-scale application in London demonstrated here can be replicated for other megacities globally to cover the needs of urban flood risk management assessments. An efficient collaboration between the insurance industry and other hazard management agencies could offer verification of the results to validate and test the estimated model depths for real rainfall events.

Further work is at hand to improve simulations in megacities by adding the storm drainage or combined sewer network. This is a major challenge, since the network data and properties are rarely available, forcing modellers to use approximations such as the UK practice of subtracting 12 mm/h from the observed rainfall. While this approach can be improved using spatially variable pipe capacity datasets (e.g., Singh et al. [71]), the high-resolution approach demonstrated here demands a similarly high-resolution and accurate representation of pressurised flows in storm drainage networks (see e.g., CityCAT capability in Bertsch et al. [21]) to account for potentially important interactions between the surface and network flows. An urgent need to establish a standardised and straightforward methodology for accurately representing sewer systems is evident, particularly in cities where datasets are scarce. This can be achieved by generating synthetic storm drains that mirror the prevailing conditions and comply with the design regulations of every country. A pioneering effort in this direction was made by Bertsch et al. [21] in a Scottish city, where they successfully calibrated and validated a systematic approach for simulating synthetic storm inlets against the existing drainage system. This approach holds promise in addressing the challenges posed by limited data availability and can significantly contribute to improving the representation of sewer systems. While this work carried out validation using observed data obtained by leveraging social media imagery for flood depth estimations during a flood event in London, this was limited to a modest number of locations due to time constraints. To obtain larger numbers of depth points, an automated approach was presented by Chaudhary et al. [72]. According to their findings, automation methods that identify objects of known dimensions, such as vehicles and individuals, could enhance accuracy as well as providing orders of magnitude larger data sets.

**Supplementary Materials:** The following supporting information can be downloaded at: https://www.mdpi.com/article/10.3390/w15193395/s1.

**Author Contributions:** Conceptualisation, C.I.; methodology, C.I.; software, C.I. and V.G.; validation, C.I. and C.K.; formal analysis, C.I.; investigation, C.I. and C.K.; resources, V.G. and C.K.; data curation, C.I.; writing—original draft preparation, C.I. and C.K.; writing—review and editing, C.I., V.G. and C.K.; visualisation, C.I.; supervision, V.G. and C.K. All authors have read and agreed to the published version of the manuscript.

**Funding:** This work was funded by the Engineering and Physical Science Research Council (EP-SRC) as part of the Centre for Doctoral Training in Water Infrastructure and Resilience (WIRe, EP/S023666/1).

**Data Availability Statement:** Not applicable.

**Acknowledgments:** The authors acknowledge support from the Willis Research Network and their encouragement in modelling flood risk in London. We thank Ferg Kilsby for his assistance with identifying the locations of the social media flood reports. We thank the three anonymous reviewers for their constructive comments.

**Conflicts of Interest:** The authors declare no conflict of interest.

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
