# Peer review of "Cloud Modelling of Property-Level Flood Exposure in Megacities"

_water, doi:10.3390/w15193395_

Round 1

Reviewer 1 Report

The paper is well written pleasant to read and as well within SI thematic. The topic is of high importance relevant to modelling approaches in high-dense urban areas such Megacities. Me suggestion for authors consideration are presented below.

Introduction: I would add more details in this section probably previous application of Citycat or other relevant information.

In chapter 2.1 there is no detailed presentation on how the surface drainage network (Figure 1 lack the surface network as input ?) is included within the model procedure even though the model is capable of using it (line 75).

Lines 120-124 You may add some references on the impact of DEM in hydrodynamic modelling (Xafoulis 2023)

Chapter 2.5: I note that only 1 hour maximum depths for different exceedances. I would expect a discussion on the catchment size of the case study and also a t justification why this duration has been selected as design base

Figure 3.2: Poor quality image. could you please add a satellite map we refer to London and it's  stunning.

Chapter 3.2: What are the downstream boundary conditions. Could you please more details herein describing the hydraulic background of the case study.

Chapter 4.3: We used to have a similar problem in our recent study (Tegos et al. 2022). Considering the the observed levels are indicative vs simulated time I suggest to provide a reasonable range (see Figure 14 by Tegos et al.)

Discussion section must be added where key points of future research will be summarized, model limitations etc.

References

Xafoulis, N.; Kontos, Y.; Farsirotou, E.; Kotsopoulos, S.; Perifanos, K.; Alamanis, N.; Dedousis, D.; Katsifarakis, K. Evaluation of Various Resolution DEMs in Flood Risk Assessment and Practical Rules for Flood Mapping in Data-Scarce Geospatial Areas: A Case Study in Thessaly, Greece. Hydrology 202310, 91 https://doi.org/10.3390/hydrology10040091

Tegos, A.; Ziogas, A.; Bellos, V.; Tzimas, A. Forensic Hydrology: A Complete Reconstruction of an Extreme Flood Event in Data-Scarce Area. Hydrology 20229, 93. https://doi.org/10.3390/hydrology905009

Author Response

Reviewer 1

The paper is well written pleasant to read and as well within SI thematic. The topic is of high importance relevant to modelling approaches in high-dense urban areas such Megacities. Me suggestion for authors consideration are presented below.

1.Introduction: I would add more details in this section probably previous application of Citycat or other relevant information.

>>We have added text in the Introduction (lines 39-46) with previous applications of CityCAT.

2.In chapter 2.1 there is no detailed presentation on how the surface drainage network (Figure 1 lack the surface network as input ?) is included within the model procedure even though the model is capable of using it (line 75).

>> We thank the reviewer for the comment and respond by saying that CityCAT can include both the surface drainage network (channels, rivers, etc.) and the sub-surface network (pipes). The surface network is automatically included (at least for high resolution implementations) through the surface DTM which will resolve any natural or artificial channels in the domain. Note, however that in London, most of the natural channels have been lost by building on top of culverted channels (apart from the River Thames and Lea).  We have changed Figure 1 by adding the sub-surface (sewer) network. For full description of the model see Glenis et al. 2018 (A fully hydrodynamic urban flood modelling system representing buildings, green space and interventions - ScienceDirect) Moreover, as mentioned in lines 204 and 205, we excluded the sub-surface drainage system from all simulations due to the limited access to the data.

3.Lines 120-124 You may add some references on the impact of DEM in hydrodynamic modelling (Xafoulis 2023)

>> We have added more text in the Introduction (lines 55-57)

4.Chapter 2.5: I note that only 1 hour maximum depths for different exceedances. I would expect a discussion on the catchment size of the case study and also a t justification why this duration has been selected as design base

We have added a sentence to justify the choice of 1 hour: “Of course, a full risk assessment should consider storms of multiple durations as well as multiple return periods (depths) to establish the overall risk which may vary across the domain as different areas will have different catchment sizes and therefore different critical durations. A comprehensive coverage of durations and return periods was not possible within this study due to computational and time constraints, so a single duration was selected for ease of analysis and comparisons with other studies.  Storm events of one hour were used for this initial study as the effective average catchment size for London is relatively small (of order 10 km2) and the majority of flooding in recent years is caused by events of around one hour duration. “  

5.Figure 3.2: Poor quality image. could you please add a satellite map we refer to London and it's  stunning.

>> We corrected the quality of the image by adding a satellite map and more details.

6.Chapter 3.2: What are the downstream boundary conditions. Could you please more details herein describing the hydraulic background of the case study.

>> We added text, and we explained that the boundary conditions of the catchments are open (line 201-202)

6.Chapter 4.3: We used to have a similar problem in our recent study (Tegos et al. 2022). Considering the the observed levels are indicative vs simulated time I suggest to provide a reasonable range (see Figure 14 by Tegos et al.)

>> Following reviewer’s suggestion, we have provide a reasonable range, and it can be seen in Figure 12   

7.Discussion section must be added where key points of future research will be summarized, model limitations etc.

>> We have changed section 5 from Conclusions to Discussion & Conclusions

Reviewer 2 Report

Dear authors

Please, read the attached file, conaining some minor comments.

Bes regards,

Author Response

Reviewer 2

  1. Lines 205-207: “While some practitioners make an allowance for this by reducing the input rainfall by e.g. 20mm, for transparency and inter-comparison we have not made any correction.”

Comment: Another choice is to reduce the rainfall intensity by the intensity corresponding to the concentration time from the IDF curve of the design frequency (according to the norms applicable when the sewage system was put into operation).

>> We would like to thank the reviewer, and we added his/her comment in the text (lines 229-232)

  1. Lines 297-299: “For clarity, the buildings at low risk have been excluded from the damage calculation by assuming that the damages are only significant for buildings identified at medium and high risk.”

Please, specify the threshold for which a building is exposed to medium or high risk.

>> The thresholds can be seen in section 2.4 – Table 2.

  1. Line 314: “Figure 7. Total numbers of inundated buildings per rainfall scenarios and per DTM resolution.”

Total numbers are presented in Figure 6, while in Figure 7 the number of buildings at high flood risk are represented. Figure 7 should probably become: “Number of buildings at high flood risk per rainfall scenarios and per DTM resolution.”

>> Following the reviewer’s comment, we have changed the caption below Figure 7.

Please, write under Figure 7, Return period as in Figure 6.

>>We have added the ‘Return period’ below Figure 7, and we would like to thank the reviewer because we missed this.

  1. Lines 333-337: “Validation against real events is a fundamental step in assessing the reliability and accuracy of hydrodynamic models…….This process …….is largely absent from commercial modelling of urban floods to date”.

This statement is debatable. Please, see below several papers using crowdsourcing data to validate hydrodynamic models:

See, L. A. (2019) Review of Citizen Science and Crowdsourcing in Applications of Pluvial Flooding. Front. Earth Sci., 7, 44. doi: 10.3389/feart.2019.00044

Blumberg, A. F., Georgas, N., Yin, L., Herrington, T. O., and Orton, P. M. (2015). Street-scale modeling of storm surge inundation along the New Jersey Hudson river waterfront. J. Atmos. Ocean. Technol. 32, 1486–1497. doi: 10.1175/JTECHD-14-00213.1

Loftis, J. D., Wang, H., Forrest, D., Rhee, S., and Nguyen, C. (2017). “Emerging flood model validation frameworks for street-level inundation modeling with storm sense,” in Proceedings - 2017 2nd International Workshop on Science of Smart City Operations and Platforms Engineering, in partnership with Global City Teams Challenge. SCOPE, Pittsburgh, PA, 13–18. doi: 10.1145/3063386. 3063764

Dinu, C.; Sîrbu, N.; Drobot, R. (2022) Delineation of the Flooded Areas in Urban Environments Based on a Simplified Approach. Appl. Sci. 2022, 12, 3174. https://doi.org/10.3390/ app12063174

>>   We believe that our statement is true in most cases, where detailed validations of surface depths, pipe pressures, and flows in and out of the sub-surface are absent. There is a reluctance among commercial model providers and consultancy users to publish such validations. However, it is also true that validations are now becoming possible due to  availability of “crowd-sourced” data, so we have changed our wording as follows, and included the (two) relevant references:

“This process ……. has largely been absent from commercial modelling of urban floods to date, but there is potential due to increased availability of flood depth data from social media and citizen science, e.g. Loftis et al., 2017 and See (2019) “

  1. Lines 453-454: “……forcing modellers to use approximations such as the UK practice of subtracting 12mm/hr from the observed rainfall.”

A value of 20 mm was mentioned earlier (lines 205-207).

>> We appreciate the reviewer’s comments and he have added a reference in lines 227-228.

Reviewer 3 Report

The manuscript entitled Cloud modelling of property level flood exposure in megacities, by C. Iliadis, V. Glenis, and C. Kilsby, presents an interesting work.

In general, the manuscript should be acceptable for publication but some serious problems must be repaired prior to publication. Some suggestions are as follows:

  1. Please follow the journal author instructions. It would be useful for the reader to follow the classical text structure (i.e. Introduction-methodology-results-discussion-conclusions). A better presentation of your results and an extensive discussion would improve your paper.
  2. Please use different terms in the “Title” and the “Keywords”.
  3. You could enrich the scientific literature.
  4. The authors could take into account the following publication: Skilodimou HD, Bathrellos GD, Alexakis DE (2021). Flood Hazard Assessment Mapping in Burned and Urban Areas. Sustainability, 13 (8): 4455, https://doi.org/ 10.3390/su13084455.
  5. Correct references in the text and the reference list according to the journal’s format. Please format the references’ list by using the correct journal abbreviations. See the following link: https://images.webofknowledge.com/images/help/WOS/A_abrvjt.html

Author Response

Reviewer 3

The manuscript entitled “Cloud modelling of property level flood exposure in megacities”, by C. Iliadis, V. Glenis, and C. Kilsby, presents an interesting work.

In general, the manuscript should be acceptable for publication but some serious problems must be repaired prior to publication. Some suggestions are as follows:

1.Please follow the journal author instructions. It would be useful for the reader to follow the classical text structure (i.e. Introduction-methodology-results-discussion-conclusions). A better presentation of your results and an extensive discussion would improve your paper.

>> We have checked up on the reviewer’s comment to see how we follow the classical text structure of the journal. We believe that although we do not use all the same words (e.g. “Results”), we are essentially very close to this structure, and our different wording and sections provide a clearer structure in the specific case of our content, so we prefer to keep the structure of our paper as it is.

2.Please use different terms in the “Title” and the “Keywords”.

>> We would like to thank the reviewer for his/her comment to use different terms for change the ‘Keywords’ . We have changed and added some terms in the keywords to add information and help readers find the paper.

Flood risk, pluvial floods, cloud computing, flood modelling, hydrodynamic model,  CityCAT, digital elevation model

3.You could enrich the scientific literature.

>> We have added text in the Introduction section.

4.The authors could take into account the following publication:

Skilodimou HD, Bathrellos GD, Alexakis DE (2021). Flood Hazard Assessment Mapping in Burned and Urban Areas. Sustainability, 13 (8): 4455, https://doi.org/ 10.3390/su13084455.

>> We thank the reviewer for drawing our attention to this interesting paper, but we don’t think it is relevant to the present work as neither the modelling methodology or scale of application are similar.  

  1. Correct references in the text and the reference list according to the journal’s format. Please format the references’ list by using the correct journal abbreviations. See the following link: https://images.webofknowledge.com/images/help/WOS/A_abrvjt.html

>> We have corrected the references in our paper.

Round 2

Reviewer 1 Report

My comments have been fully addressed and therefore I would recommend to publish it as it is. Well deserved.

Author Response

My comments have been fully addressed and therefore I would recommend to publish it as it is. Well deserved.

>> We would like to thank the reviewer for their previous constructive comments and suggestions.

Reviewer 3 Report

The manuscript entitled Cloud modelling of property level flood exposure in megacities, by C. Iliadis, V. Glenis, and C. Kilsby, presents an interesting work.

In general, the manuscript should be acceptable for publication but some problems must be repaired prior to publication. Some suggestions are as follows:

  1. Please follow the journal author instructions. It would be useful for the reader to follow the classical text structure (i.e. Introduction-methodology-results-discussion-conclusions). A better presentation of your results and an extensive discussion would improve your paper.

Author Response

The manuscript entitled “Cloud modelling of property level flood exposure in megacities”, by C. Iliadis, V. Glenis, and C. Kilsby, presents an interesting work.

In general, the manuscript should be acceptable for publication but some problems must be repaired prior to publication. Some suggestions are as follows:

1.Please follow the journal author instructions. It would be useful for the reader to follow the classical text structure (i.e. Introduction-methodology-results-discussion-conclusions). A better presentation of your results and an extensive discussion would improve your paper.

>> We believe we have already addressed these comments in the first round.  The comments appear to be generic and do not refer to any specific instances, which would help us understand where the reviewer thinks we are falling short. We respectfully suggest that the reviewer may not have read and understood the significant changes we made after the first round of reviews.

Nonetheless, we have checked again to see how we follow the classical text structure of the journal and how extensive our discussion is.

  1. We believe the structure is fit for purpose and clear to readers for the specific content of the paper. However, to make it even clearer, and even more “standard”, we have changed the section titles to reflect the referee’s first suggestion and the paper now has a section titled: “Results - Flood risk in London”.
  2. In response to the comment on “extensive discussion”, we have added text in the Discussion and Conclusions section to improve our paper as follows:

“While this work has carried out validation using observed data obtained by leveraging social media imagery for flood depth estimations during a flood event in London, this was limited to a modest number of locations due to time constraints. To obtain larger numbers of depth points, an automated approach such as presented by Chaudhary et al. [73]. According to their findings, automation methods that identify objects of known dimensions, such as vehicles and individuals, could enhance accuracy as well as providing orders of magnitude larger data sets.”

Round 3

Reviewer 3 Report

The manuscript entitled “Cloud modelling of property level flood exposure in megacities” presents an improved and good work.

The manuscript should be acceptable for publication in the present form.